# RETRACTED: Fe_3_O_4_ Nanoparticles for Complex Targeted Delivery and Boron Neutron Capture Therapy

**DOI:** 10.3390/nano9040494

**Published:** 2019-03-31

**Authors:** Kanat Dukenbayev, Ilya V. Korolkov, Daria I. Tishkevich, Artem L. Kozlovskiy, Sergey V. Trukhanov, Yevgeniy G. Gorin, Elena E. Shumskaya, Egor Y. Kaniukov, Denis A. Vinnik, Maxim V. Zdorovets, Marina Anisovich, Alex V. Trukhanov, Daniele Tosi, Carlo Molardi

**Affiliations:** 1School of Engineering, Nazarbayev University, 010000 Nur-Sultan, Kazakhstan; 2The Institute of Nuclear Physics, 050032 Almaty, Kazakhstan; 3L.N. Gumilyov Eurasian National University, 010008 Nur-Sultan, Kazakhstan; 4Laboratory of Magnetic Films Physics, Cryogenic Research Department, Scientific-Practical Materials Research Centre, National Academy of Sciences of Belarus, 220072 Minsk, Belarus; 5Laboratory of Single crystal growth, South Ural State University, 454080 Chelyabinsk, Russia; 6Department of Electronic Materials Technology, National University of Science and Technology MISiS, 119049 Moscow, Russia; 7Ural Federal University named after the First President of Russia B.N. Yeltsin, 620075 Yekaterinburg, Russia; 8Republican Unitary Enterprise “Scientific-Practical Centre of Hygiene”, 220012 Minsk, Belarus

**Keywords:** magnetic nanoparticles, iron oxide, surface functionalization, APTMS, carborane

## Abstract

Magnetic Fe_3_O_4_ nanoparticles (NPs) and their surface modification with therapeutic substances are of great interest, especially drug delivery for cancer therapy, including boron-neutron capture therapy (BNCT). In this paper, we present the results of boron-rich compound (carborane borate) attachment to previously aminated by (3-aminopropyl)-trimethoxysilane (APTMS) iron oxide NPs. Fourier transform infrared spectroscopy with Attenuated total reflectance accessory (ATR-FTIR) and energy-dispersive X-ray analysis confirmed the change of the element content of NPs after modification and formation of new bonds between Fe_3_O_4_ NPs and the attached molecules. Transmission (TEM) and scanning electron microscopy (SEM) showed Fe_3_O_4_ NPs’ average size of 18.9 nm. Phase parameters were studied by powder X-ray diffraction (XRD), and the magnetic behavior of Fe_3_O_4_ NPs was elucidated by Mössbauer spectroscopy. The colloidal and chemical stability of NPs was studied using simulated body fluid (phosphate buffer—PBS). Modified NPs have shown excellent stability in PBS (pH = 7.4), characterized by XRD, Mössbauer spectroscopy, and dynamic light scattering (DLS). Biocompatibility was evaluated in-vitro using cultured mouse embryonic fibroblasts (MEFs). The results show us an increasing of IC_50_ from 0.110 mg/mL for Fe_3_O_4_ NPs to 0.405 mg/mL for Fe_3_O_4_-Carborane NPs. The obtained data confirm the biocompatibility and stability of synthesized NPs and the potential to use them in BNCT.

## 1. Introduction

Magnetic iron oxide nanoparticles (NPs), such as magnetite (Fe_3_O_4_) and maghemite (γ-Fe_2_O_3_), have been found and applied in a wide range of biomedical applications [1,2,3,4,5,6,7,8,9], including magnetic resonance imaging, magnetic hyperthermia, cancer therapy, and drug delivery; they also find applications in catalysis [10,11,12] and in magnetic separation [13,14]. Among other metal nanoparticles, iron oxide nanoparticles have been more extensively studied in clinical practice [15]. A wide range of Fe_3_O_4_ NPs’ applications are derived from unique properties, such as the superparamagnetism, high surface area, and nanometer-level size [8]. At the same time, iron oxide NPs, stand-alone, have a tendency for agglomeration, low stability in solutions, and insufficient biocompatibility. The application of iron oxide NPs in medicine reveals one more issue: An intravenous delivery agent should be eliminated from the blood and not cause accumulation in the organism, which can lead to side effects. Arami et al. [16] showed that iron oxide NPs are safe and non-toxic with a median lethal dose of 300–600 mgFe/kg. Zang et al. [17] reported that the accumulation and elimination of iron oxide NPs were found on a sample of zebrafish, and as a result, magnetic nanoparticles were eliminated from 86% to 100% by 24 days. Weissleder et al. [18] studied the toxic effect on rats and beagle dogs, revealing that no toxicity was detected in animals that received 3000 mmol Fe/kg overall. Furthermore, various materials, such as silane-based compounds, metals, polymers, and fatty and amino acids, have been used for coating and stabilizing the surface of iron oxide NPs in order to reduce toxicity and increase biocompatibility [19,20,21]. Among other materials, silane-based compounds are most promising since they have high biocompatibility, stability, low toxicity, low cost, and high prospects for functionality [22,23,24,25]. Also, the modification of iron oxide NPs by silanes with functional groups can lead to an easy attachment of payloads. One of these payloads can be a boron-rich compound for potential use as drug in boron neutron capture therapy of cancer (BNCT).

BNCT is a promising method of cancer treatment based on selective accumulation of boron-rich compounds inside the cancer cells and its neutron irradiation. Boron compounds and neutrons collide and cause atomic fission, producing α-Ray radiation; as a result, cancer cells are damaged from the inside [26]. BNCT was primarily used for head and neck cancer treatment [27,28], but improvements of the method turned it into a promising therapy for soft tissues, like breast cancer [29], glioblastoma [30], melanomas of the skin, and liver metastases [31]. However, BNCT has limitations, which are the high concentration of thermal neutrons in the order of 10 n/cm^2^ and high concentration of boron-10 in the tumor (20–35 μg per 1 g of tissue) [32,33]. These drawbacks can be improved by the immobilization of different boron-rich compounds, such as carboranes, into carriers such as micelles [34], monoclonic antibodies [35], liposomes [36,37], carbon nanotubes [38], gold nanoparticles [39], and magnetic nanostructures [40,41]. The main requirements for the use of nanostructures as containers for targeted drug delivery are the possibility of transporting drugs directly to the affected tissues or organs, reducing side effects, and the non-toxicity and stability of the carriers [26]. Among them, magnetic iron oxide NPs are preferable due to the potential possibility of drug targeting into each specific site of the disease [42]. Zhu et al. [40] studied the effective method of carborane (10 atoms per molecule) immobilization on Fe_3_O_4_ NPs by catalytic azide-alkyne cycloaddition reactions. However, this method is complicated (consisting of three stages) and required expensive reagents. Moreover, it is necessary to increase the concentration of boron content for successful application in BNCT.

In this paper, we present the result of Fe_3_O_4_ NPs modification with (3-aminopropyl)-trimethoxysilane (APTMS) and subsequent simple attachment with carborane borate, containing 21 boron atoms. The stability of this system was elucidated by using a model-solution with simulated body fluid (pH = 7.4). The biocompatibility was evaluated in vitro using cultured mouse embryonic fibroblasts.

## 2. Materials and Methods

The structural and magnetic properties of magnetic NPs and the dynamics of their degradation in phosphate buffer were studied, as well as the method of modifying particles with an organosilicon compound, in order to prevent degradation of the NPs’ surface. The possibility of immobilizing carboranes on a functionalized surface was studied. Structural changes and their effects on the magnetite properties of the resulting composite particles were determined. To identify the toxicity of the magnetic particles themselves, composites, and the attached carboranes, toxicological studies were carried out on an MTT (methyltetrazolium test) assay with different NPs using mouse embryonic fibroblasts (MEFs). The scheme of the proposed research is presented in Figure 1.

### 2.1. Synthesis and Modification of Fe_3_O_4_ NPs

NPs based on iron oxide were obtained by co-precipitation of a mixture of iron (II) chloride and (III) chloride with the addition of ammonium hydroxide. This reaction can be represented as follows:FeCl_2_ + 2FeCl_3_ + 8NH_3_ · H_2_O→Fe_3_O_4_ + 8NH_4_Cl + 4H_2_O.

2 M FeCl_2_ was dissolved in 2 M HCl, 1 M FeCl_3_ in a molar ratio of 1:2. 50 mL of NH_4_OH (0.7 M) was added dropwise through the funnel to the resulting solution over a period of 5–10 min, while stirring with a magnetic stirrer. After the synthesis, the resulting samples were washed in an ultrasonic bath and dried at 50 °C for 24 h.

### 2.2. Amine Functionalization of NPs and Carborane Immobilization

Amine functionalization of the NPs’ surface was done by adding 1 mL of (3-aminopropyl) trimethoxysilane (APTMS) with a concentration of 20 mM in ethanol to Fe_3_O_4_ NPs. The reaction mixture was ultrasonicated for 30 min and kept in these solutions for 24 h.

After amine functionalization, the samples were dried and washed in ethanol. For the binding, carborane with an aminated surface of the powder of the NPs was added to 0.1 M solution of carborane borate, synthesized as in [43] in ethanol, and the reaction mixture was ultrasonicated for 30 min and was kept in a shaker for 24 h. After completing the reaction, the samples were washed with ethanol and dried.

### 2.3. Methods of Characterization

Structural and compositional analysis was performed with either a JEOL JEM 2100 LaB_6_ (JEOL Ltd, Akishima, Tokyo, Japan), or with a ARM200F transmission electron microscope TEM, JEOL Ltd, Akishima, Tokyo, Japan) operated at 200 kV.

X-ray diffraction analysis was carried out on a D8 ADVANCE ECO diffractometer (Karlsruhe, Germany) using CuKα radiation. In order to identify the phases and study the crystal structure, the software, BrukerAXSDIFFRAC.EVAv.4.2 (Bruker, Karlsruhe, Germany), and the international ICDD PDF-2 database were used.

FTIR (Fourier transform infrared spectroscopy) spectra were recorded on an Agilent Technologies Cary 600 Series FTIR Spectrometer with Single Reflection Diamond Attenuated total reflectance (ATR) accessory (GladiATR, PIKE, Fitchburg, WI, USA) to study chemical group shifts before and after NPs’ modification. Measurements were taken in the range of 400 to 4000 cm^−1^. All spectra (40 scans at a 4.0 cm^−1^ resolution) were recorded at 21–25 °C. Spectral analysis was conducted by using Agilent Resolution Pro.

The Mössbauer studies were carried out using a spectrometer, MS1104Em (Chernogolovka, Russia). The ^57^Co nuclei in the Rh matrix were used as a source. The Mössbauer spectrometer was calibrated at room temperature with a α-Fe standard.

The investigation of the magnetic properties was carried out using the vibrational magnetometer system (VSM, Liquid Helium Free High Field Measurement System “Cryogenic Ltd.”, London, UK). The measurements were implemented by the induction method, through a determination of the induced electromotive force of the induction in signal coils by a magnetized sample oscillating with a definite frequency at magnetic field of H = ±20,000 Oe at a 300 K temperature.

### 2.4. Stability of Fe_3_O_4_ NPs

PBS solution was prepared from PBS tablets (Sigma-Aldrich, St.Louis, MO, USA) to obtain a 137 mM NaCl, 2.7 mM KCl, and 10 mM phosphate buffer solution (pH 7.4 at 25 °C). Samples were immersed into PBS solution for 1, 5, and 10 days, and the sample was then washed with deionized water, dried, and weighted. Control was performed by using XRD, Mössbauer spectroscopy, and DLS analysis (Malvern Panalytical, Malvern, UK).

### 2.5. Cytotoxicity

The biocompatibility of Fe_3_O_4_ NPs, Fe_3_O_4_-Aminated NPs and Fe_3_O_4_-Carborane NPs was studied in vitro by 3-(4,5-dimethylthiazol-2-yl)-2,5-diphenyltetrazolium bromide MTT assay with different NPs concentration from 0.005 to 0.6 mg/mL using mouse embryonic fibroblasts (MEFs). All tests were carried out at temperature of 20–22 °C, humidity of 32–44%, and atmospheric pressure of 733–745 mmHg using a centrifuge OPN-3, microscope Axiovert 40C and 40, thermohygrometer Testo-608-H-1 (Testo, Moscow, Russia).

#### 2.5.1. Preparation of Cells

The material for the MEFs culture was obtained by disaggregating the skin and muscle tissue of 12–14-day mouse embryos using 0.25% trypsin solution at 37 °C for 30 min [44]. After the removal of trypsin, the sediment was suspended in growth medium (90% of Dulbecco’s Modified Eagle Medium (DMEM), 10% of fetal calf serum with the addition of antibiotics). The suspension was passed through a filter (cell 0.3 × 0.3 mm). After counting in the Goryaev chamber, cells were sown in vials with a seed density of 100–300 thousand per 1 cm^2^ of growth surface and cultured in a growth medium for 5–7 days. The number of cells of different morphotypes was counted in accordance with the classification of Bayreuter [45].

#### 2.5.2. Cell Cytotoxicity Assay (MTT)

The MTT is an assay to identify metabolic disorders, namely, the dysfunction of mitochondria, reflecting the effect on cell viability. Cells were grown in a CO_2_ incubator (Herra Cell) at 37 °C, 5% CO_2_, 80% relative humidity on 96-well plates (seed concentration—50–70 thousand cells/mL). Samples of NPs dissolved in fetal bovine serum (Sigma, St.Louis, MO, USA) were added to the wells with adherent cells (second day of cultivation). After a 24-h exposure of the samples, total cell mitochondrial dehydrogenase activity in each well was measured photometrically in the methyltetrazolium test (MTT). This test is based on the ability of living metabolically active cells to convert the tetrazoline salt (MTS) into formazan, which is soluble in the culture medium. Thus, the absorption of formazan is directly proportional to the number of viable cells in the culture. CellTiter 96^®^ AQueous One Solution Cell Proliferation Assay (MTS), Promega kit, was used for MTT. To measure the absorption of formazan, the cells were incubated with MTS for 4 h in a thermostat, and the measurement of the absorption of formazan at λ = 490 nm was performed on an automatic microplate photometer EIX808, BioTek Instruments Inc., Winooski, VT, USA. The toxicity of nanoparticles was assessed by IC_50_.

## 3. Results

### 3.1. Synthesis of Fe_3_O_4_ Nanoparticles

Fe_3_O_4_ NPs were synthesized using a mixture of iron (II) and (III) chloride with the addition of ammonium hydroxide [46]. As a result, spherical NPs with sizes up to 20 nm, according to the TEM images (Figure 2a), were produced. The same sizes varied from 8.64 to 28 nm and the average size of 18.9 nm was verified by DLS analysis (Figure 2b).

According to X-ray diffraction analysis (Figure 2c), the synthesized nanoparticles are polycrystalline of magnetite Fe_3_O_4_ with a cubic structure; the lattice parameter, a = 8.24467 ± 0.00013 Å, differs from the reference value (PDF—01-075-9673, Fe_3_O_4_—Cubic, Fd-3m(227) a = 8.34800 Å). The results of the X-ray phase analysis on the formation of magnetite nanoparticles are confirmed by the results of the Mössbauer spectroscopy. Unlike the X-ray diffraction patterns of nanoparticles, which are characterized by low-intensity broadened diffraction peaks that can correspond to both magnetite and maghemite due to the similarity of the crystal structure and main diffraction maxima, the analysis of the Mössbauer spectra makes it possible to accurately determine the magnitude of the hyperfine magnetic field, which is different for different iron oxide phases.

The Mössbauer spectra of all the synthesized samples (Figure 2d) were taken at room temperature in the range of −10/+10 mm/s. The spectra correspond to a Zeeman sextet with broadened resonance lines, characteristic of Fe_3_O_4_. The Mössbauer data were analyzed using the method of restoration of the hyperfine parameters’ distributions of the Mössbauer spectrum taking into account a priori information about the samples. A model consisting of two sextets characteristic of tetrahedral (A-phase) and octohedral (B-phase) positions of iron atoms in the structure and a quadrupole doublet was used as a model for the analysis. According to literature data for nanoscale magnetite, the presence of two sextets is due to Fe^3+^ in the positions of the A sublattice (spinel) and intermediate valence states of Fe^2.5+^ in the positions of the B sublattice. According to the obtained data, the asymmetric shape and the broadening of the spectral lines is due to the presence of regions of disorder in the structure. The magnitudes of the hyperfine magnetic fields for the A and B sublattices are 472.5 ± 2.5 kOe and 442 ± 4.1 kOe, respectively, which is in good agreement with the literature data on the magnitudes of the hyperfine fields for magnetite nanostructures [47,48].

The ratio of the contributions of the partial spectra for the studied nanoparticles was Fe^3+^_tetra_/Fe^3+,2+^_oct_ ≈ 0.57, which is close enough to the reference value of −0.5 characteristic of stoichiometric magnetite [49,50,51]. The deviation of the obtained ratio of contributions from the reference value is due to the presence in the structure of vacancy defects and regions of disorder that arise during the synthesis. The presence of a low-intensity quadrupole doublet in the structure, which is characteristic of cationic defects in the structure, indicates the presence of regions of disorder or impurity paramagnetic inclusions in the crystal structure, the presence of which is due to synthesis processes.

The appearance of the hysteresis loop for the synthesized nanoparticles is characteristic of nanoscale soft ferromagnetic materials. The hysteresis loop (inserts to Figure 2e has a form typical for a superparamagnetic material. Increased fragments of loops (in the range of B = ±1000 Oe) demonstrate the magnetic characteristics: Coercivity is about 10 Oe, saturation magnetization is 68 emu/g, which is typical for Fe_3_O_4_ magnetic NPs [52].

The appearance of the hysteresis loop for the synthesized nanoparticles is characteristic of nanoscale soft ferromagnetic materials. The hysteresis loop (insets of Figure 2e has the typical form of a superparamagnetic material. The increased fragments of loops (in the range of B = ±1000 Oe) demonstrate the magnetic characteristics: Coercivity is about 10 Oe, saturation magnetization is 68 emu/g, which is typical for Fe_3_O_4_ magnetic NPs [52,53].

### 3.2. Modification of Fe_3_O_4_ Nanoparticles

Amine functionalization of the NPs’ surface was provided with organosilicon compound in order to prevent degradation and to functionalize the surface of NPs. A schematic representation of the functionalization and characteristics of the modified NPs is presented in Figure 3a.

The formation of the bonds of the modifying agent with the surface of the NPs was proven by ATR-FTIR spectroscopy (Figure 3b). The ATR-FTIR spectrum of the initial Fe_3_O_4_ NPs is characterized by absorption at 3500–3000 cm^−1^ (OH groups), at 1614 cm^−1^ related to O–H bending vibrations combined with Fe atoms, and at 544 and 399 cm^−1^ corresponding to Fe–O. Aminated Fe_3_O_4_ NPs are characterized by new peaks at 1602 cm^−1^ (NH_2_), at 1223 and 994 cm^−1^ (Si–C), and at 1050 and 1112 cm^−1^ (Si–O–Si).

X-ray diffraction patterns of the modified samples (Figure 3c) in the range of 2θ = 15–25° show the presence of peaks corresponding to APTMS (hexagonal type). An increase in the lattice parameter of a = 8.30458 Å for the modified samples indicates the interaction of these compounds (APTMS) with the crystalline structure of NPs by replacing iron or oxygen atoms in the lattice sites or introducing them into interstitial sites [54]. Particle size distribution after amination is presented in Figure 3d. There is a slight increase in the average size compared with the initial Fe_3_O_4_ NPs 

EDX (energy-dispersive X-ray, Hitachi, Chiyoda, Tokyo, Japan) analysis was performed in order to obtain the element ratio variations after modification with APTMS and are presented in Table 1. It was found that amination with APTMS led to the appearance of Si and N in the atomic content of NPs.

As for the initial NPs, the spectra of the modified ones (Figure 3e) correspond to a Zeeman sextet with broadened resonance lines, characteristic of Fe_3_O_4_, and a quadrupole doublet, characteristic of the paramagnetic state of Fe^2+^ and Fe^3+^. The decrease in the hyperfine field (Figure 3f) for the modified samples by APTMS to 466.8 ± 2.1 kOe and 432 ± 1.7 kOe is due to the modification of the NPs’ surface, the change in the structural bonds, and the introduction of –Si, –O, and –NH_3_ into the interstice of the crystal lattice, thus leading to its change [49,55,56].

Increased fragments (inserts to Figure 3g) of loops (in the range of B = ±1000 Oe) demonstrate a slight decrease in the magnetic characteristics in comparison with the initial Fe_3_O_4_ NPs, caused by changing the present nonmagnetic faze on the magnetic NPs’ surface: Coercivity is about 6 Oe, saturation magnetization is 62 emu/g, and this is close to the magnetic parameters of Fe_3_O_4_ NPs, as indicated in [57].

Thus, the proposed method for the functionalization of magnetic nanoparticles allows the surface of nanostructured samples to be aminated, which was confirmed with the results of the FTIR, XRD, and EDX analysis and Mössbauer spectroscopy. The slight changes in structure lead to an insignificant increase in particle size, and the appearance of non-magnetic inclusions that make a slight contribution to the magnetic characteristics of the resulting composite material. Due to the presence of amino groups on the surface of the samples, it becomes possible to attach payloads to the surface of magnetic nanoparticles, which is a promising lead to be used as magnetic nanoscale carriers for targeted drug delivery, bioseparation, and other biological applications. The obtained results are in good agreement with previous work on the functionalization of the surface of oxide nanoparticles and their potential use in biomedicine and targeted drug delivery [55,58,59].

Magnetic media used for biological purposes should have the following properties: Maintain their structural, chemical, and physical properties over time, while being in biological environments; not have a direct toxicological effect on biological objects (cells); the procedure for the addition of drugs to magnetic NPs should be simple and should not affect the particles themselves and the attached substances.

In order to confirm the possibility of using the obtained magnetic NSs in bio-medical applications, the following studies were conducted:1Dynamics of changes in structural properties over a model environment, similar in properties to biological fluids (PBS solution with an acidity of 7.4 t = 25 ° C).2Immobilization of carboranes on the functionalized amino groups’ surface of the nanoparticles.3Study of the toxicological properties of the magnetic carrier, magnetic carrier with a functionalized surface, and samples with immobilized carboranes.

### 3.3. Stability of Fe_3_O_4_ NPs in PBS Solution

One of the important parameters of the use of NPs as magnetic carriers for targeted drug delivery is their resistance to degradation in various media. PBS solution with an acidity of 7.4 is widely used as simulated body fluid for degradation testing of magnetic carriers [60,61,62]. The most common mechanism of degradation is amorphization and the subsequent destruction of the crystal structure and the deterioration of the magnetic characteristics of NPs. It is necessary to know the exact time of the change in the properties of NPs in order to determine the optimal area and time of effective application.

Figure 4 shows the changes in the main diffraction peaks of the initial and modified Fe_3_O_4_ NPs, depending on the time of degradation. The time range of measurements was 10 days, control points: 1, 5, and 10 days.

According to the presented data, no new diffraction peaks appear in the diffractograms of the samples, which indicates the absence of phase transformations in the process of degradation. The result of the calculations of the structural parameters (relative change of lattice parameter (a_0_ − a)/a_0_, crystalline size, crystallinity) as well as in the size of NPs, studied by the DLS method, and changes in the structure, investigated based on the Mössbauer spectra, are presented in Table 2. By approximating the lines of the diffractogram with the necessary number of symmetric pseudo-Voigt functions, the width of the recorded lines at half of their height (FWHM, full-width half maximum) was measured, which allowed us to characterize the perfection of the crystal structure and estimate the degree of crystallinity [63,64]:
(1)
PV(x,x0,η,bL,bG,A)=A[(1−η)·G(x,x0,bG)+η·L(x,x0,bL)],G(x,x0,bG)=exp[−(x−x0)22bG2],L(x,x0,bL)=11+(x−x0bL)2

where: *x*—variable corresponding to the angle of reflection, 2θ; *x*_0_—sets the position of the maximum function; *η*—specific proportion of the Lorentz function; *A*—normalization factor; *b_G_* and *b_L_*—Gauss function parameters, *G*(*x*, *x*_0_, *b_G_*) and Lorentz *L*(*x*, *x*_0_, *b_L_*). As a criterion for such compliance, we can use the standard deviation, the minimum value of which, when these parameters vary, corresponds to their optimal set:
(2)
σ=∑i(PV(x,x0,η,bL,bG,A)i−Ii)2n

where: 
(PV(x,x0,η,bL,bG,A)i
—pseudo-Voigt function value; *I_i_*—value of experimental intensity; *i*—reflex profile point number; *n*—number of points in the profile. Figure 5 shows the changes in the Mössbauer spectra of the initial Fe_3_O_4_ NPs and aminated Fe_3_O_4_ NPs during degradation.

The main changes are associated with the decrease in the intensity and shape of the diffraction peaks, as well as a small shift of the peaks that indicates a change in the parameters of the crystal lattice and the size of crystallites. The largest changes in the intensities are observed for the initial unmodified Fe_3_O_4_ NPs. Less degradation of modified nanoparticles is due to the presence of organosilicon compounds coating the NPs’ surface, which actively interacts with the PBS medium less and reduces the rate of degradation.

According to the data, the largest increase in the lattice parameter is also observed for unmodified Fe_3_O_4_ NPs, which is caused by the processes of the introduction of oxygen ions into the internode of the crystal lattice and its subsequent swelling [65,66]. Herewith, the modification of the surface leads to a smaller value of the parameter change, which, as mentioned above, is due to the low penetration of oxygen and hydrogen ions through the modified layer.

An increase in the average crystallite size is observed for modified nanoparticles, which is caused by the modification of the crystal structure. The crystallite size was estimated using the Debye-Scherer method [67]. Degradation for 1 day led to a sharp increase in the average crystallite size of initial Fe_3_O_4_ NPs, which may be due to the incorporation of oxygen into the nanoparticle structure, in which changing the main crystallographic characteristics leads to a decrease in the degree of perfection of the crystal structure due to the formation of amorphous inclusions [68,69]. Fe_3_O_4_ modified with organosilicon compound shows smaller changes in the average crystallite size.

A change in the crystal structure as a result of a decrease in the degree of perfection and the formation of amorphous inclusions can lead to magnetic disordering and a subsequent change in the magnitude of the hyperfine magnetic field. The results of a change in the magnitude of the hyperfine magnetic field, determined by Mössbauer spectroscopy, depending on the time of degradation, show that an increase in the degradation time leads to a change in the structural characteristics and the formation of amorphous inclusions in the structure, which in turn leads to a decrease in the magnitude of the hyperfine magnetic field [70,71]. This is due to the formation of cationic vacancies and impurity inclusions in the structure, as well as the disordering of the magnetic texture [72].

Thus, modified Fe_3_O_4_ NPs with APTMS showed great stability in PBS solution during 1–10 days. XRD analysis and Mössbauer spectroscopy show minor changes in the crystal structure associated with an increase of the crystallites’ sizes and a decrease in the degree of perfection of the crystal parameters. At the same time, the products of degradation, such as FeOOH or Fe_3_(PO_4_)_2_, as they were previously reported in [61,62] were not detected.

Excellent colloidal stability is associated with the stabilization of the system by electrostatic and steric repulsion. For the initial Fe_3_O_4_ NPs, the average hydrodynamic size of NPs increased from 18.9 to 27.9 nm, while for the Fe_3_O_4_-Aminated NPs, it increased from 22.4 to 26.2 nm, during 10 days when kept in PBS solution. The largest changes in hydrodynamic size occurred in the initial Fe_3_O_4_ NPs (9 nm), the smallest changes in the Fe_3_O_4_-Aminated NPs. The obtained data are in good agreement with the data on XRD analysis.

### 3.4. Immobilization of Carborane Borate

Attachment of carborane was carried out through the ionic interaction of carborane borate with amino groups of the modified Fe_3_O_4_ NPs (Figure 6a).

Based on the FTIR spectra (Figure 6b), the mechanism of attaching carborane can be observed. The reaction of the NH_2_ bond with carborane borate led to the observation of new peaks at 2625 cm^−1^ (B–H of carborane core) and the appearance of NH_3_^+^ group vibrations at 1521 cm^−1^. The presented results are in good correlation with previously published work [43,73]. Based on the EDX spectra, the presence of B is confirmed in the FTIR results (Table 3).

The size of NPs with carborane, estimated with DLS (Figure 6c), is slightly increased to 24.3 nm from 19.4 nm (initial NPs).

According to the X-ray diffraction analysis (Figure 6d), an increase in the lattice parameter of a = 8.30786 Å (APTMS + Carborane) for the modified samples indicates the interaction of these compounds with the crystalline structure of NPs by replacing iron or oxygen atoms in the lattice sites or introducing them into interstitial sites. X-ray diffraction patterns of the modified samples in the range of 2θ = 15–25° show the presence of peaks corresponding to APTMS (hexagonal type) and carborane borate (inset of Figure 6d). This indicates the formation of stable chemical bonds between the coatings and the structure of the nanoparticles.

The Mössbauer spectra (Figure 6e) correspond to a Zeeman sextet with broadened resonance lines, characteristic of Fe_3_O_4_, and an extended quadrupole doublet, characteristic of the paramagnetic state of Fe^2+^ and Fe^3+^. The broadened inhomogeneous lines of the spectra, as well as the presence of a quadrupole doublet in the spectrum, indicate an increase of magnetic disordering in the structure and the absence of a selected magnetic texture. The decrease in the hyperfine field (Figure 6f) for the modified samples of 452.1 ± 1.7 and 427± 2.1 kOe for samples with attached carborane borate is due to the modification of the NPs’ surface [74,75].

The hysteresis loop (Figure 6g) for the samples of NPs + APTMS + Carboranes is still apparent, but corresponds to soft ferromagnetic materials. The magnetic characteristics are: Coercivity is about 70 Oe, saturation magnetization is 31 emu/g. The change in the size and structure of NPs functionalized by carborane and the properties of the modified surface influenced the magnetic properties of NPs. These changes are possibly associated with an increase in the dipole–dipole interaction between the particles, leading to rather large changes in the coercivity. An increase in the amount of the nonmagnetic phase (carboranes and amorphous phase) leads to a decrease in the specific saturation magnetization, thus remaining at a sufficiently high value.

Thus, we have demonstrated the possibility of immobilizing carboranes on the surface of magnetic NPs functionalized by amino groups. Following the proposed technique, stable chemical bonds are formed, ensuring the presence of carborane on the surface. It is important to note that the proposed method induces some changes in the structure of NPs and their surfaces, leading to a significant change in the structural and magnetic parameters. These changes, however, do not lead to the loss of the target properties of magnetic carriers. These methods, as well as the method of magnetic NPs, could be proposed for targeted delivery of carboranes using a magnetic field to a tumor in boron capture therapy.

### 3.5. Cytotoxicity

As preliminary in vitro studies, the biocompatibility of modified and non-modified Fe_3_O_4_ NPs was tested by an MTT assay using MEFs. Cells were incubated with NP concentrations from 0.005 mg/mL to 0.6 mg/mL for 24 h. Figure 7a shows that NPs did not present any toxicity for a concentration up to 0.015 mg/mL. IC_50_ was determined graphically from the curve presented in Figure 7b, and the results are presented in Table 4. The IC_50_ for the initial Fe_3_O_4_ NPs is 0.110 mg/mL, and for Fe_3_O_4_-aminated NPs it is 0.091 mg/mL and for Fe_3_O_4_-Carboranes NPs it is 0.405 mg/mL. Fe_3_O_4_ NPs’ toxicity depends on a variety of factors, including the dose, exposure time, colloidal and chemical properties of the surface, the NPs’ hydrodynamic diameter, and NPs’ protein interactions [8].

The increased cytotoxicity of NPs coated with APTMS can be explained by the fact that the surface of Fe_3_O_4_-Aminated NPs contains positively charged NH_2_-groups that can electrostatically interact with negatively charged cell membranes [54,76]. Subsequent binding of carborane neutralizes the amino group, which leads to a decrease in the cytotoxicity of Fe_3_O_4_-Carboranes NPs.

It is noteworthy that a dose of about 0.5 mg/mL for any compound is the maximum dose allowed for testing in vitro, above which the effects are not specific. As can be seen from the obtained data, the half inhibitory concentration for Fe_3_O_4_-Carboranes NPs approaches this value. This suggests that Fe_3_O_4_-Carboranes NPs have the lowest cytotoxicity. Their cytotoxicity is possibly associated with a purely mechanical effect of the high concentration, which prevents adhesion and, accordingly, cell growth in culture.

## 4. Conclusions

Magnetic Fe_3_O_4_ NPs with an average size of 18.9 nm were synthesized and modified with APTMS. NPs were characterized using FTIR, TEM, XRD, DLS, Mössbauer spectroscopy, and VSM. The technique of successful functionalization of the NPs’ surface by organosilicon compound (APTMS), thanks to which the functional properties of the NPs do not change, but the possibility of the immobilization of payloads to functional amino groups of the modified surface, was confirmed.

To demonstrate the possibility of using synthesized nanoparticles as magnetic carriers for the purpose of targeted delivery, studies of the degradation of initial nanoparticles and nanoparticles functionalized with APTMS were carried out, the possibility of immobilizing carborane borates (target substances used for boron capture therapy) on a functionalized surface was carried out, and toxicological studies of both magnetic carriers and their modifications were performed.

It was shown that magnetic NPs are prone to a slight change in their structure even in magnetic neutral medium (PBS). However, the use of APTMS as a surface functionalizing agent also contributed to reducing degradation.

Carborane borate with 21 boron atoms per molecule was successfully attached to the amino group of nanoparticles modified with APTMS for potential use in BNCT. Despite some changes in the structural and magnetic properties, when using the proposed technique for immobilizing carboranes, the functional properties of the NPs and the stable chemical bonds formed by the carboranes with the surface suggest its applicability.

The NPs Fe_3_O_4_ biocompatibility test showed an increase in the IC_50_ from 0.110 mg/mL for NPs Fe_3_O_4_ to 0.4005 mg/mL for Fe_3_O_4_-carborane NPs. The data obtained confirm the biocompatibility and stability of the synthesized Fe_3_O_4_-carborane NPs and identify the possibility of further testing as carriers for the targeted delivery of boron to cancer cells.

## Figures and Tables

**Figure 1 nanomaterials-09-00494-f001:** Schematic representation of the study.

**Figure 2 nanomaterials-09-00494-f002:** Characteristics of the initial magnetic nanoparticals: (**a**) TEM image; (**b**) size distribution, (**c**) XRD diffractogram; (**d**) Mössbauer spectra and the distribution of the hyperfine magnetic field; (**e**) hysteresis loop.

**Figure 3 nanomaterials-09-00494-f003:** Characteristics of the functionalized magnetic nanoparticles: (**a**) Schematic representation of the modification of magnetic nanoparticals by organosilicon compound (APTMS); (**b**) FTIR spectra, (**c**) XRD diffractogram; (**d**) size distribution; (**e**) Mössbauer spectra; (**f**) distribution of the hyperfine magnetic field; (**g**) hysteresis loop.

**Figure 4 nanomaterials-09-00494-f004:** The changes in the main diffraction peaks of (**a**) the initial and (**b**) modified Fe_3_O_4_ NPs.

**Figure 5 nanomaterials-09-00494-f005:** Mössbauer spectra of the studied nanoparticles.

**Figure 6 nanomaterials-09-00494-f006:** Characteristics of the system magnetic nanoparticles + APTMS + Carboranes. (**a**) Schematic representation of carborane derivatives immobilization; (**b**) FTIR spectra; (**c**) size distribution; (**d**) XRD diffractogram; (**e**) Mössbauer spectra; (**f**) the distribution of the hyperfine magnetic field; (**g**) hysteresis loop.

**Figure 7 nanomaterials-09-00494-f007:** Cytotoxic effect of modified and non-modified Fe_3_O_4_ NPs (**a**) on cultured mouse embryonic fibroblasts and its nonlinear curve fitting (**b**).

**Table 1 nanomaterials-09-00494-t001:** Element ratio variations by using the EDX method of magnetic nanoparticles and after modification with APTMS.

Sample	Atomic Content, %	[NH_2_], µM/g
N	Fe	Si	O
Fe_3_O_4_ NPs	-	43.1	-	56.9	-
Fe_3_O_4_-Aminated NPs	1.0	36.0	2.3	60.7	86.57

**Table 2 nanomaterials-09-00494-t002:** Changes in the structure of magnetic nanoparticles and after modification with APTMS occurring during degradation in PBS solution.

Analysis	Parameter	Fe_3_O_4_	Fe_3_O_4_-Aminated NPs
Degradation, days	0	1	5	10	0	1	5	10
XRD	Relative change of lattice parameter, % (a_0_ = 8.34800 Å, PDF—01-075-9673)	0.074	0.089	0.097	0.103	0.021	0.035	0.047	0.048
Crystalline size, nm	10.3 ± 0.7	12.1 ± 0.6	12.2 ± 0.9	12.9 ± 0.3	12.8 ± 0.8	12.5 ± 0.7	12.7 ± 0.4	12.8 ± 1.1
Crystallinity, %	79.4 ± 3.1	77.9 ± 1.5	67.8 ± 2.4	57.6 ± 2.3	80.1 ± 1.8	79.5 ± 1.5	75.9 ± 1.4	67.1 ± 2.2
DLS	Average size of particles, nm	18.9 ± 3.2	19.66 ± 3.14	26.52 ± 4.1	27.88 ± 3.9	21.8 ± 3.56	21.4 ± 3.7	23.46 ± 4.23	26.22 ± 4.15
Mossbauer	Hyperfine field, kOe	A-site	472.9 ± 2.5	453.2 ± 3.5	434.1 ± 4.6	421.6 ± 2.7	466.8 ± 8.6	452.2 ± 4.3	441.3 ± 4.7	422.2 ± 4.3
B-site	442 ± 4.1	434 ± 3.2	432 ± 4.4	417 ± 2.2	432 ± 1.7	426 ± 2.2	421 ± 2.4	418 ± 2.3

**Table 3 nanomaterials-09-00494-t003:** Element ratio variations by using the EDX method of Fe_3_O_4_ nanoparticles with carborane.

Sample	Atomic Content, %
B	N	Fe	Si	O
Fe_3_O_4_-Carborane NPs	11.4	2.0	33.5	12.2	40.9

**Table 4 nanomaterials-09-00494-t004:** Inhibitory concentration (IC_50_) for samples using MEFs.

No.	Sample	IC_50_, mg/mL
1	Fe_3_O_4_ NPs	0.110
2	Fe_3_O_4_-Aminated NPs	0.091
3	Fe_3_O_4_-Carboranes NPs	0.405

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
