# Peer review of "RETRACTED: Fe3O4 Nanoparticles for Complex Targeted Delivery and Boron Neutron Capture Therapy"

_nanomaterials, 2019, doi:10.3390/nano9040494_

Round 1
Reviewer 1 Report
The paper faces a very important clinical application and the research approach is very complete and is performed very accurately, presenting many different and complementary experimental investigations and explaining into details their rationale and conclusions.
My only suggestion would be that of defining more precisely whether the concentration in boron finally obtained in the tumor is that expected for effective BNCT, based on the cross section of the corresponding reaction.
The text should be carefully checked, some of the mis-spellings are below:
line 20: including ; 40: including; 52: is a promising; 60: to reach ; 80: arrow missing in the chemical formula; 180: functionalize the surface;217: fase; 316: based on.
Author Response
1) My only suggestion would be that of defining more precisely whether the concentration in boron finally obtained in the tumor is that expected for effective BNCT, based on the cross section of the corresponding reaction.
In this paper we studied methods of immobilization of carboranes on magnetic NPs and its cytotoxicity effect. We reached 23% of boron per Fe3O4 NP. In our next research we will carry out in vivo test on boron distribution in tissue of such systems without external magnet, and with external magnet. We can expect that this concentration in boron in the tumor will be enough for effective BNCT since the authors (Zhu, Y.; Lin, Y.; Zhu, Y.Z.; Lu, J.; Maguire, J.A.; Hosmane, N.S. Boron Drug Delivery via Encapsulated Magnetic Nanocomposites: A New Approach for BNCT in Cancer Treatment. J. Nanomater. 2010, 2010, 1–8) reach around 50 µg/g boron in tumor, although they load NPs with only around 10% of boron per NPs.
According to the requirements to BNCT, necessary concentration of born should be 20-35 μg/g.
2) The text should be carefully checked, some of the mis-spellings are below: line 20: including ; 40: including; 52: is a promising; 60: to reach ; 80: arrow missing in the chemical formula; 180: functionalize the surface; 217: fase; 316: based on.
All mistakes were corrected.
3) Moderate English changes required.
English was improved.
Reviewer 2 Report
This manuscript presents a trivial procedure to synthesize and characterize iron oxide nanoparticles. May be the unique novelty of the paper is the attempt το functionalize the NPs with carborane borate. In general, the paper is written in bad English, and contains many assumptions not supported by the experimental data, faulty judgments, absent and/or wrong reference citations, arbitrary data interpretations. In the attached file several comments are given in detail.

Author Response
Reviewer #2
1) Extensive editing of English language and style required.
English errors were corrected. Colored in yellow in the manuscript.
2) Does the introduction provide sufficient background and include all relevant references? – NO.
Introduction was expanded.
3) Hematite not.
This mistake was corrected. Hematite was deleted
4) This is not true. And there is not such a statement in Ref. 15.
The ref 15 have been corrected and replaced to Koda, J.; Venook, A.; Walser, E.; Goodwin, S. A multicenter, phase I/II trial of hepatic intra-arterial delivery of doxorubicin hydrochloride adsorbed to magnetic targeted carriers in patients with hepatocellular carcinoma. Eur. J. Cancer 2002, 38, S18.
5) prevent functionalization?
This mistake was corrected
6) There is not such a synthesis in Ref. 27.
Ref. 27. was corrected
7) There is not such a classification mentioned in Ref. 29.
Ref. 29. was corrected
8) PBS is not the appropriate medium to study the stability of magnetite in biological environments.
PBS solution can be used as medium to simulate the degradation behavior of iron oxide NPs since the medium ion concentration is quite similar to the blood plasma one. This is also supported by follow refernces:
1. Tolouei, R.; Harrison, J.; Paternoster, C.; Turgeon, S.; Chevallier, P.; Mantovani, D. The use of multiple pseudo-physiological solutions to simulate the degradation behavior of pure iron as a metallic resorbable implant: a surface-characterization study. Phys. Chem. Chem. Phys. 2016, 18, 19637–19646.
2. da Silva, G.; Marciello, M.; del Puerto Morales, M.; Serna, C.; Vargas, M.; Ronconi, C.; Costo, R. Studies of the Colloidal Properties of Superparamagnetic Iron Oxide Nanoparticles Functionalized with Platinum Complexes in Aqueous and PBS Buffer Media. J. Braz. Chem. Soc. 2016, 28, 731–739.
3. Li, X.; Lachmanski, L.; Safi, S.; Sene, S.; Serre, C.; Grenèche, J.M.; Zhang, J.; Gref, R. New insights into the degradation mechanism of metal-organic frameworks drug carriers. Sci. Rep. 2017, 7, 13142.
9) Which is?
(PDF – 01-075-9673, Fe3O4 – Cubic, Fd-3m(227) a=8.34800 Å).
10) The X-ray diffractogramm is inaccurate, with wide peaks. maghemite would be the case also. Phase characterization baesd on this XRD plot is not adequate of a superparamagnetic material.
The results of X-ray phase analysis on the formation of magnetite nanoparticles are confirmed by the results of Mössbauer spectroscopy. Unlike X-ray diffraction patterns of nanoparticles, which are characterized by low-intensity broadened diffraction peaks, which can correspond to both magnetite and maghemite due to the similarity of the crystal structure and main diffraction maxima, the analysis of the Mössbauer spectra makes it possible to accurately determine the magnitude of the hyperfine magnetic field, which is different for different iron oxide phases. According to the obtained data, the magnitude of the hyperfine magnetic field for the nanoparticles is 472.9 ± 1.2 kOe, which corresponds to the structure of magnetite in the A and B positions of the sublattice [47,48]. While for the structure of maghemite, the characteristic value of the fields is above 500 kOe [49]. Moreover, according to [50–52], the formation of magnetite nanoparticles is characteristic of the chemical synthesis of nanoparticles without thermal annealing.
47. Iyengar, S.J.; Joy, M.; Ghosh, C.K.; Dey, S.; Kotnala, R.K.; Ghosh, S. Magnetic, X-ray and Mössbauer studies on magnetite/maghemite core–shell nanostructures fabricated through an aqueous route. RSC Adv. 2014, 4, 64919–64929.
48. Ghosh, R.; Pradhan, L.; Devi, Y.P.; Meena, S.S.; Tewari, R.; Kumar, A.; Sharma, S.; Gajbhiye, N.S.; Vatsa, R.K.; Pandey, B.N.; et al. Induction heating studies of Fe3O4 magnetic nanoparticles capped with oleic acid and polyethylene glycol for hyperthermia. J. Mater. Chem. 2011, 21, 13388.
49. da Costa, G.M.; De Grave, E.; Vandenberghe, R.E. Mössbauer studies of magnetite and Al-substituted maghemites. Hyperfine Interact. 1998, 117, 207–243.
50. Shouheng, S.; Zeng, H. Size-Controlled Synthesis of Magnetite Nanoparticles. 2002.
51. Xu, Z.; Shen, C.; Hou, Y.; Gao, H.; Sun, S. Oleylamine as Both Reducing Agent and Stabilizer in a Facile Synthesis of Magnetite Nanoparticles. Chem. Mater. 2009, 21, 1778–1780.
52. Visalakshi, G.; Venkateswaran, G.; Kulshreshtha, S.K.; Moorthy, P.N. Compositional characteristics of magnetite synthesised from aqueous solutions at temperatures upto 523K. Mater. Res. Bull. 1993, 28, 829–836.
11) Reference value? Standard deviation?
Wang, Shige, et al. "Aminopropyltriethoxysilane-mediated surface functionalization of hydroxyapatite nanoparticles: synthesis, characterization, and in vitro toxicity assay." International journal of nanomedicine 6 (2011): 3449.
12) This is an arbitrary conclusion.
Obtained results are in a good agreement with the work on the functionalization of the surface of oxide nanoparticles and their potential use in the biomedicine and targeted drug delivery [54–56].
54. Sawatzky, G.A.; Coey, J.M.D.; Morrish, A.H. Mössbauer Study of Electron Hopping in the Octahedral Sites of Fe 3 O 4. J. Appl. Phys. 1969, 40, 1402–1403.
55. Chowdhuri, A.R.; Bhattacharya, D.; Sahu, S.K. Magnetic nanoscale metal organic frameworks for potential targeted anticancer drug delivery, imaging and as an MRI contrast agent. Dalt. Trans. 2016, 45, 2963–2973.
56. Dobson, J. Magnetic nanoparticles for drug delivery. Drug Dev. Res. 2006, 67, 55–60.13) Fig.1 Metal oxide NPs, not metal NPs.
13) Fig.1 Metal oxide NPs, not metal NPs.
This mistake was corrected.
14) Same comment as previous.
Increased fragments (inserts to Fig. 3g) of loops (in the range B = ± 1000 Oe) demonstrate a slight decrease in magnetic characteristics in comparison with initial Fe3O4 NPs, caused by changing the present nonmagnetic faze on the magnetic NPs surface: coercivity is about 6 Oe, saturation magnetization is 62 emu/g, and close to magnetic parameters of Fe3O4 NPs, indicated in [57].
57. Arsalani, N.; Fattahi, H.; Nazarpoor, M. Synthesis and characterization of PVP-functionalized superparamagnetic Fe3O4 nanoparticles as an MRI contrast agent. Express Polym. Lett. 2010, 4, 329–338.
14) Row 342 What is the magnetization value with respect to the magnetite mass? 31 emu/g with taking into account the organic mass is not a reliable value.
Mass of functionalized and coated NPs were not estimated. That’s why its not possible to say mass of drugs. Here an example of decreasing saturation magnetization around estimatied level 10.3144/expresspolymlett.2010.42 10.1371/journal.pone.0158084.g008
All other answers on Reviewer #2 comments are colored in yellow in the revised manuscript.

Round 2
Reviewer 2 Report
The authors tried to comply in many of my initial remarks but not in all. There are still serious faulty judgments. The main problem remains the interpretation of the Moessbauer spectra. The authors use only one sextet to fit the data and in the same time they refer and compare to Refs. using two sextets. The misuse of irrelevant refs and/or claims which are not to be found in those references is very problematic. Overall, as I wrote in my first review, the novelty of this manuscript is limited and together with the faulty judgments does to justify publication in a journal like Nanomaterials.
see my comments in the attached file.

Author Response
We thank the Referee for their interest in our work and for helpful comments that will improve the manuscript and we have tried to do our best to respond to the points raised. The Referee has brought up some good points and we appreciate the opportunity to clarify our research objectives and results. As indicated below, we have checked all comments provided by the Referee and have made necessary changes accordingly to their indications.
Bellow we attached the answers of reviewer’s comments and in text we highlighted in yellow our corrections.
Response to Reviewer
Comment 1. - The authors tried to comply in many of my initial remarks but not in all. There are still serious faulty judgments. The main problem remains the interpretation of the Moessbauer spectra. The authors use only one sextet to fit the data and in the same time they refer and compare to Refs. using two sextets. The misuse of irrelevant refs and/or claims which are not to be found in those references is very problematic.
- Obviously, you mean "superparamagnetic nanoparticles" due to their small size? There is not such a thing as "paramagnetic state of Fe2+ and Fe3+" in magnetite.
- In both references [47,48] two sextets are used to fit the A and B positions as always in spinel type structures and not one as is the case in this manuscript. Moreover, there is not such a value for the hyperfine field (472.9) in those references as the authors claim.
- You can not compare between hyperfine field values of bulk materials and those of nanoparticles.
Response:
The authors are grateful to the referee for the valuable comment on the analysis of the Mössbauer data. The spectra were analyzed using a model with two sextets and one quadrupole doublet. A detailed description is provided below.
The Mössbauer spectra of all the synthesized samples were taken at room temperature in the range of -10/+10 mm/s. The spectra correspond to a Zeeman sextet with broadened resonance lines, characteristic of Fe3O4. The Mössbauer data were analyzed using method of restoration of the hyperfine parameters distributions of the Mössbauer spectrum taking into account a priori information about the samples. A model consisting of two sextets characteristic of tetrahedral (A-phase) and octohedral (B-phase) positions of iron atoms in the structure and a quadrupole doublet was used as a model for the analysis. According to literature data for nanoscale magnetite, the presence of two sextets is due to Fe3+ in the positions of the A sublattice (spinel) and intermediate valence states of Fe2.5+ in the positions of the B sublattice. According to the obtained data, the asymmetric shape and the broadening of the spectral lines is due to the presence of regions of disorder in the structure. The magnitudes of the hyperfine magnetic fields for the A and B sublattices are 472.5 ± 2.5 kOe and 442 ± 4.1 kOe, respectively, which is in a good agreement with the literature data on the magnitudes of the hyperfine fields for magnetite nanostructures [1,2].
The ratio of the contributions of the partial spectra for the studied nanoparticles was Fe3+tetra/Fe3+,2+oct ≈ 0.57, which is close enough to the reference value - 0.5 characteristic of stoichiometric magnetite [3–5]. The deviation of the obtained ratio of contributions from the reference value is due to the presence in the structure of vacancy defects and regions of disorder that arise during the synthesis. The presence of a low-intensity quadrupole doublet in the structure, which is characteristic of cationic defects in the structure, indicates the presence of regions of disorder or impurity paramagnetic inclusions in the crystal structure, the presence of which is due to synthesis processes.
Comment 2. What is the experimental error of the unit cell constant a?
Response: The lattice parameter was determined from the main diffraction peaks. The error is 0.00013Å.
Comment 3. Overall, as I wrote in my first review, the novelty of this manuscript is limited and together with the faulty judgments does to justify publication in a journal like Nanomaterials.
Response: The novelty of this manuscript is an encapsulation of boron-rich compound that consist of 21 boron atom per molecule on previously aminated magnetic iron oxide nanoparticles with a high load. Selective accumulation of boron in the tumor cells produces a substantial therapy ratio during boron neutron capture therapy of cancer (BNCT). Consequently, improvements in boron delivery vehicles will dramatically improve BNCT effectiveness. The most important problem of BNСT is the synthesis of boron-organic compounds, capable to selective accumulation in tumorous tissues. It was calculated that at maximum flux of epithermal neutrons at 1013 n/сm2, the boron-10 content should achieve 109 atoms per cells or 20-35 µg/g under the condition of equal distribution in cancer cells. Encapsulation of molecule consisting of 55 % of boron atoms on magnetic nanostructures can potentially enhance effectiveness of BNCT. In addition, unlike other supported drugs, it is not necessary that the attached boron species be released from the particle supports for them to be effective. Methods of synthesis and modification are simple, easy and cheap, which makes this technology available for widespread use in the future.
Herein, we report our preliminary results on the modification of magnetic iron oxide NPs with a high load of carborane borate compound along with a preliminary study of their biocompatibility and stability. The results of the project can be potentially used in biomedicine.
References
1. Si, S.; Kotal, A.; Mandal, T.K.; Giri, S.; Hiroyuki Nakamura, A.; Kohara, T. Size-Controlled Synthesis of Magnetite Nanoparticles in the Presence of Polyelectrolytes. 2004.
2. Daou, T.J.; Pourroy, G.; Bégin-Colin, S.; Grenèche, J.M.; Ulhaq-Bouillet, C.; Legaré, P.; Bernhardt, P.; Leuvrey, C.; Rogez, G. Hydrothermal Synthesis of Monodisperse Magnetite Nanoparticles. 2006.
3. Ghosh, R.; Pradhan, L.; Devi, Y.P.; Meena, S.S.; Tewari, R.; Kumar, A.; Sharma, S.; Gajbhiye, N.S.; Vatsa, R.K.; Pandey, B.N.; et al. Induction heating studies of Fe3O4 magnetic nanoparticles capped with oleic acid and polyethylene glycol for hyperthermia. J. Mater. Chem. 2011, 21, 13388.
4. Yurenya, A.; Nikitin, A.; Garanina, A.; Gabbasov, R.; Polikarpov, M.; Cherepanov, V.; Chuev, M.; Majouga, A.; Panchenko, V. Synthesis and Mössbauer study of 57Fe-based nanoparticles biodegradation in living cells. J. Magn. Magn. Mater. 2019, 474, 337–342.
5. Lyubutin, I.S.; Lin, C.R.; Korzhetskiy, Y. V.; Dmitrieva, T. V.; Chiang, R.K. Mössbauer spectroscopy and magnetic properties of hematite/magnetite nanocomposites. J. Appl. Phys. 2009, 106, 034311.